# Deeper Learning By Doing: Integrating Hands-On Research Projects Into A Machine Learning Course

**Sebastian Raschka** [1]

## Abstract

Machine learning has seen a vast increase of interest in recent years, along with an abundance of learning resources. While conventional lectures provide students with important information and knowledge, we also believe that additional project-based learning components can motivate students to engage in topics more deeply. In addition to incorporating project-based learning in our courses, we aim to develop project-based learning components aligned with real-world tasks, including experimental design and execution, report writing, oral presentation, and peer-reviewing. This paper describes the organization of our project-based machine learning courses with a particular emphasis on the class project components and shares our resources with instructors who would like to include similar elements in their courses.

## 1. Motivation

Interests in machine learning (ML) and deep learning (DL) have been increasing in recent years. Similarly, the number of learning resources, including textbooks, blogs, online courses, and video tutorials, is growing rapidly as well. This is a great development, and one might say that getting into ML has never been easier.

However, we believe that while the process of *absorbing* knowledge from various resources is necessary, it is not sufficient for becoming a successful ML researcher or practitioner. Anecdotal evidence from online learning communities suggests that adopting an *experimental mindset* can accelerate learning (Osmulski, 2021). Moreover, analyses by Headden and McKay assert that "a sense of control over the work" is an essential aspect for motivating and engaging students in learning (Headden & McKay, 2015). How can we foster such an experimental mindset and engage

students? While we cannot answer this definitively, in this paper, we describe our DL course featuring project-based learning components, where students work on original questions and research topics that interest them.

Three years ago, we began designing ML and DL courses with substantial student project components, including an original research proposal, conference paper-style project report, oral class presentation, and paper peer-review. We have adopted and refined this approach throughout teaching six ML and DL courses. While similar project-based elements were used in different ML and DL courses, this paper will only focus on the latest DL course.

Based on anonymous surveys, the project-based learning components were, without exceptions, very well received by the students. In addition, we found that it was effective in fostering interaction and collaboration among students and offering students opportunities to practice essential communication skills. This paper outlines our latest project-based course format as well as some of the lessons learned.

## 2. Overall Course Design

This section briefly outlines the overall course and lecture design to provide the broader context for the project-based learning components described in more detail in Section 3.

### 2.1. Target Audience

The course is listed as an elective course for statistics and data science majors and is thus aimed at senior undergraduate students. Programming and scientific computing experience is highly recommended, but prior ML knowledge is not required.

### 2.2. Lecture Topics

Being intended as an introductory course that exposes students to all major areas of DL, we introduce students to the core concepts of DL via face-to-face lectures over the course of 15 weeks. The course covers all major areas of DL, including backpropagation, multi-layer perceptrons, convolutional neural networks for image data, recurrent neural networks and transformers for text data, and varia-

[1]Department of Statistics, University of Wisconsin-Madison, Madison, WI, USA. Correspondence to: Sebastian Raschka <sraschka@wisc.edu>.

*Proceedings of the 2nd Teaching in Machine Learning Workshop*, PMLR, 2021. Copyright 2021 by the author(s).

tional autoencoders and generative adversarial networks for generating image data.

We omit a detailed lecture topic list due to page limit constraints, but interested readers can find a list of lecture topics in our supplementary material[1].

### 2.3. Implementing Algorithms from Scratch and Using Libraries

While the general DL topics and concepts are taught in a conventional lecture format, using a tablet to augment presentation slides with rich annotations, we prepare and discuss full code examples as demonstrations for each topic.

We agree with Schiendorfer et al. (2021) that it could be beneficial to expose students to "from scratch" implementations in addition to teaching how to use established libraries. These implementations have pedagogical value since they serve as an additional "language" (in addition to drawings and mathematics) to describe algorithms. In addition, being familiar with coding algorithms from scratch can help students with being able to implement and experiment with their ideas more readily.

However, implementing algorithms from scratch is both inefficient and error-prone. Hence, we believe that it is in the students' best interest to balance from-scratch implementations and using existing libraries. For example, after presenting students with the essential conceptual and mathematical details, we teach students how to implement a logistic regression classifier trained with stochastic gradient descent[2]. Then, we show students how the same can be achieved with PyTorch (Paszke et al., 2019) and its automatic differentiation capabilities[3]. We think that empowering students to implement algorithms from scratch but also showing more reliable and convenient tools motivates and demystifies the latter.

### 2.4. Student Work and Evaluation

Besides attending the lectures, students are presented with weekly quizzes, homework (approximately every two weeks), a midterm exam, and the class project components, which will be detailed in Section 3. The project structure and timeline are summarized in Figure 1.

While it appears that students are presented with a substan-

---

[1] https://github.com/rasbt/
ecml-teaching-ml-2021/blob/main/
lecture-topics.md
[2] https://github.com/rasbt/
ecml-teaching-ml-2021/blob/main/nbs/logreg_
from-scratch.ipynb
[3] https://github.com/rasbt/
ecml-teaching-ml-2021/blob/main/nbs/logreg_
pytorch.ipynb

tial workload at first glance, the students reported that the workload in this course indeed presents an average course load for a three credit point course.

The weekly self-assessment quizzes are multiple-choice, multiple dropdown, numerical, and multiple answer style questions that test the students' current understanding of the course material. These quizzes constitute only a small percentage of the total grade but help incentivize students to keep up with the lecture material before and after the midterm exam. There is no final exam in this course as we found that it adds unnecessary stress when the students prepare the deliverables for the project-based components towards the end of the semester.

Through the homework assignments, students learn to implement and apply core concepts learned in the lectures. In contrast to the weekly quizzes, the homework assignments are coding-based. Since DL code can be very verbose and include a lot of boilerplate code, students are provided with skeleton code where they only need to fill in key parts. We encourage students to reuse lecture and homework code in their class projects.

## 3. Project-based Learning Components

Considering that ML is primarily a very applied field, we believe that ML courses can benefit from project-based learning components. In this regard, we designed a course with the class project as a major component, where the sum of its components constitutes half of the total grade. The individual components consist of (1) a project proposal, (2) a report, (3) an oral presentation, and (4) peer-review. Overall, this process aims to mimic the lifecycle of a real-world ML project from conception to completion.

### 3.1. Forming Project Groups

To provide students with sufficient time to work on their project proposals (Figure 1), project groups should ideally be formed as soon as possible, within the first weeks of the semester. An added benefit of creating project groups early is that the project groups can also function as study groups.

In the absence of strong evidence in favor of a particular group size, we initially considered group sizes of 2-4 students. In a classroom of 72 students, we preferred sizes of 3-4 to reduce the total number of groups to improve instructor support and extend the per-group presentation time for the oral in-class presentations at the end of the semester. Furthermore, following advice from research into different group sizes (Apedoe et al., 2012), we took advice from teacher impressions suggesting that "Groups of 3 worked best." Upon request, we allow students to select their group partners and assign remaining slots randomly.

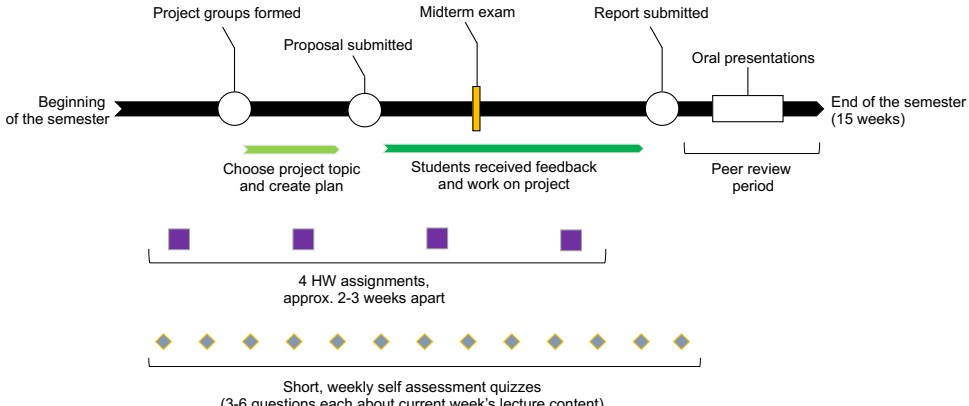

*Figure 1.* Summary and timeline of the student deliverables throughout the semester.

## 3.2. Project Proposal

The project proposal is a short 2-3 page document outlining the project plans. Students receive total points if all sections in the template[4] are completed because the proposal's main intention is to provide instructors with a formal outline of the student's plan for feedback. The proposal's due date is set to approximately 2-3 weeks after the project groups are formed such that students have enough time remaining in the semester to work on the project itself.

A particular challenge is that students are asked to propose a DL project without having been exposed to the breadth of topics covered in class. While this is unavoidable for practical reasons, we recommend sharing interesting and diverse examples and applications of DL with students early in the semester to help students to help with choosing the topic and defining the approximate scope. In addition, we found that providing examples of anonymized project proposals from previous semesters can make this task less daunting.

In retrospect, while some groups found the project conceptualization more challenging than others, we never encountered a case where students couldn't find a project they were interested in working on. In addition, projects that students worked on in the past were very diverse. For example, projects included convolutional neural network based self-driving cars, COVID-19 detection, and trading card game classification. We included anonymized example reports in the supplementary materials[5].

---

[4]https://github.com/rasbt/
ecml-teaching-ml-2021/tree/main/
proposal-template

[5]https://github.com/rasbt/
ecml-teaching-ml-2021/tree/main/
project-examples

## 3.3. Project Report

While we realize that in the real world, papers and paper sections can be flexible and diverse, we aim to create a universal rubric that can be applied fairly to all projects for grading[6]. For this purpose, we adopted the CVPR conference template for the report and defined the following sections: Abstract, Introduction, Related Work, Proposed Method, Experiments, Results and Discussion, Conclusions, and Contributions. We share this template and provide more details about the section contents in the supplementary material[7] along with anonymized example reports from previous semesters[8].

To keep the writing and reviewing efforts realistic and manageable, the require students to stay within 6-8 pages excluding references. In addition, we provide students with the aforementioned report rubric to assist their writing efforts. We recommend students to use Overleaf[9] (free tier) as it provides the best collaborative writing experience for LaTeX papers.

## 3.4. Project Presentation

At the end of the semester, students present their projects in class. Due to practical reasons, the presentation length is capped at 8 minutes, and presentations are split across 3 separate lecture days (8 presentations per lecture days).

---

[6]https://github.com/rasbt/
ecml-teaching-ml-2021/blob/main/rubrics/
report-rubric.md

[7]https://github.com/rasbt/
ecml-teaching-ml-2021/tree/main/
report-template

[8]https://github.com/rasbt/
ecml-teaching-ml-2021/tree/main/
project-examples

[9]https://www.overleaf.com/

To further incentivize attendance, the presentation order is randomized (announced at the beginning of each class), and we give bonus points for attendance. We track attendance through voting sheets, where students are asked to vote for their preferred candidates for the *Best Oral Presentation*, *Most Creative Project*, and *Best Visualizations* awards.

### 3.5. Peer Review

Students are expected to review two project reports and presentations from other groups. This is a single-blind setting where reviewers remain anonymous to the project group members. We provide code to facilitate this peer review assignment[10]. Given that project groups consist of three students each, 5-6 reviewers were assigned to each project. The reviewer scores were averaged, and outliers were removed at the instructor's discretion. To make the presentation and report assessments as fair as possible, the reviewers received detailed rubrics to follow[11]. (These rubrics were shared several weeks before the report due date such that students could use those as additional guidance during the writing process.) In addition, peer-reviewers received points for each submitted review to incentivize complete and timely submissions. The instructors curated the peer reviews, and constructive feedback was shared with the students alongside the instructors' feedback.

We found that this peer-review process worked exceptionally well, and students appreciated this experience. Also, in addition to the instructor feedback, the peer reviews create an additional opportunity for students to receive feedback and being exposed to different perspectives, which can help with improving their work. A downside of this approach is that feedback could sometimes be overly harsh, for instance, assigning zero points for related work when a report included such related work in the introduction section but omitted/removed the related work section (originally part of the report template) itself. We are thinking of future versions of the rubric to allow more flexibility and bonus points for exceptionally well-done sections.

### 3.6. Switching from In-Person to All-Virtual

In 2020 and 2021, the COVID-19 pandemic required switching the course to an all-virtual format. While this was a new experience for both instructors and students, we could transition all aspects of the course to a virtual environment without making major changes to the course design. In-person lectures were replaced by virtual lectures and recordings to accommodate students in different time zones. We made

accommodations during the group assignment such that students in similar time zones were working together. While students use collaborative tools in a non-virtual semester (e.g., GitHub for code sharing, Overleaf for collaborative writing, and OneDrive for general file sharing), students used conferencing software for virtual meetings, and similar to the in-class presentations, the student presentations were pre-recorded such that students could view them at their convenience. We found that the possibility of pre-recording their talks helped students overcome nervousness related to speaking in front of an audience, and we are considering offering this as an option in future in-person semesters.

### 3.7. Reception

While we have no formal way (for example, via AB testing) to assess the success of the project-based learning, we are under the impression that it was worthwhile. Generally, the course was very well received (averaging a 4.8/5.0 overall course rating in recent semesters). In anonymous class surveys conducted by the college, students included the following comments: "Project is somewhat challenging but very meaningful;" "One of my favorite courses I've taken in college;" "I enjoyed this course and I really enjoyed the final project."

The project-based components of this course may suggest that the primary goal of ML research is to produce publications. However, we observed that the process of gaining experience with applying predictive or generative models to real-world data and getting feedback on how they can improve is what students value the most. Thus, based on our observations, we think that we successfully convey that producing publications is not the centerpiece of ML research.

Moreover, from personal communications with the instructors, students mentioned that the class project was a helpful resume component when interviewing for internships or jobs.

## 4. Conclusion

This paper has presented a DL course that includes substantial project-based learning components and shared the templates and rubrics we created and refined in previous semesters. Without exception, student feedback has been unilaterally supportive of the class project in recent semesters. We noticed that most students were very motivated to research DL topics beyond the scope of this course. While project-based learning components may create extra work for the instructors, we think seeing the creative outcomes is very rewarding. Moreover, project-based learning provides additional opportunities for meaningful student collaborations and interactions.

---

[10]https://github.com/rasbt/
ecml-teaching-ml-2021/tree/main/
review-assignment

[11]https://github.com/rasbt/
ecml-teaching-ml-2021/tree/main/rubrics

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
