# OpenReview forum: "Deeper Learning By Doing: Integrating Hands-On Research Projects Into A Machine Learning Course"
_ecmlpkdd.org/ECMLPKDD/2021/Workshop/TeachML — TeachML 2021_

### Official Review · Reviewer_7M12 · 2021-07-14
**Very clear method report, perhaps too much publication-focused**

**Rating:** 8
**Confidence:** 5

**Review:**

The paper is very clearly written and follows the submission guidelines exactly.

The authors report on "project-based learning" used in a deep learning course. The authors make use of the known fact that applying knowledge is a more effective learning strategy than only absorbing it. In their approach, the course is structured as an "end-to-end" research project, including an "original research proposal, conference paper-style project report, oral class presentation, and paper peer-review" (by students). This is very interesting and goes beyond assignments that, for instance, require students to replicate published results.

The authors mention an important point, which is implementing selected basic ML methods from scratch for educational purposes but then using libraries (e.g. PyTorch) in production. Other important components of their course are weekly self-assessment quizzes and coding homework, as well as the absence of a final exam. Instead the paper (using a common skeleton structure to help grading) and oral presentations are graded.

One aspect that could be improved: The course seems to paint a paper-focused picture, which may give students the impression that the main goal of machine learning research (applied or method development) is to produce publications. The authors could have commented on whether they observed this message to have been suggested to students as a result of the course.

---

### Official Review · Reviewer_bS3J · 2021-07-16
**A project-based machine learning course**

**Rating:** 8
**Confidence:** 5

**Review:**

In this work authors describe a machine learning course with a strong focus on machine learning projects to be done by the students.

The paper is clear and well presented. What is interesting in this paper is that students are required to do peer review on the projects of theirs colleagues making it an interesting approach. The authors have reported adaptability to online setup due to covid-19 pandemic. Authors have reported the student reception which has been positive as a statement of considering student opinions.

It would be interesting to extend the Lecture Topics section at least one paragraph, it is very broad explained without enough detail about the topics covered.

---

### Decision · Program_Chairs · 2021-07-21

**Decision:**

Accept

**Comment:**

Congratulations! The reviewers agree that this paper should be accepted.

Camera-ready version is due August 18, 2021. As you prepare the camera ready version, please take the reviewers comments into consideration.

We look forward to your participation at the workshop on September 13, 2021. We invite you also to join us for the satellite event on September 08, 2021. Schedules for both the workshop and the satellite event will be forthcoming.